# Ethical review of COVID-19 research in the Netherlands; a mixed-method evaluation among medical research ethics committees and investigators

R. IJkema *, M. J. P. A. Janssens, J. A. M. van der Post, C. M. Licht

Medical Research Ethics Committee, VU medical center, Amsterdam, The Netherlands

* r.ijkema@amsterdamumc.nl

## Abstract

### Background

During the beginning of the COVID-19 pandemic there was an urgent need for accelerated review of COVID-19 research by Medical Research Ethics Committees (MRECs). In the Netherlands this led to the implementation of so-called 'fast-track-review-procedures' (FTRPs) to enable a swift start of urgent and relevant research. The objective of this study is to evaluate FTRPs of MRECs in the Netherlands during the COVID-19 pandemic and to compare them with the regular review procedures (RRPs).

### Methods and findings

An explanatory sequential mixed method study was conducted. Online questionnaires and four group interviews were conducted among MREC representatives and investigators of COVID-19 research. In addition, data from a national research registration system was requested. Main outcome measures are differences in timelines, quality of the review and satisfaction between FTRPs and RRPs. The total number of review days was shorter in FTRP (median 10.5) compared to RRPs (median 98.0). Review days attributable to the MRECs also declined in FTRPs (median 8.0 versus 50.0). This shortening can be explained by installing ad hoc (sub)committees, full priority given to COVID-19 research, regular research put on hold, online review meetings and administrative leniency. The shorter time-lines did not affect the perceived quality of the review and ethical and legal aspects were not weighted differently. Both MREC representatives and investigators were generally satisfied with the review of COVID-19 research. Weaknesses identified were the lack of overview of COVID-19 research and central collaboration and coordination, the delay of review of regular research, and limited reachability of secretariats.

### Conclusions

This study shows that accelerated review is feasible during emergency situations. We did not find evidence that review quality was compromised and both investigators and MRECs

**Data Availability Statement:** a. Datasets questionnaires: a. MRECs b. investigators. These datasets were de-identified and uploaded to the

DANS data repository: https://doi.org/10.17026/dans-z2r-hnfa b. Datasets obtained from the national registration system of the CCMO. Publicly sharing these data is restricted by the CCMO. Requests can be sent to metc@vumc.nl. c. Interview transcripts and analysis files. Since it is not possible to fully anonymize these data because of potentially/indirectly identifying clauses, it is not ethical to share these publicly. Data requests can be sent to metc@vumc.nl.

**Funding:** The author(s) received no specific funding for this work.

**Competing interests:** The authors have declared that no competing interests exist.

were content with the FTRP. To improve future medical ethical review during pandemic situations and beyond, distinguishing main and side issues, working digitally, and (inter)national collaboration and coordination are important.

## Introduction

Since the beginning of 2020, like the rest of the world, the Netherlands has been confronted with the COVID-19 pandemic caused by the SARS-CoV-2 virus [1]. Mortality, severe morbidity, the unfamiliarity with the virus and the lack of knowledge about effective treatment, detection, spread and vaccines, has led to large-scale and urgent medical scientific research initiatives in order to fill these knowledge gaps. Many of the Medical Research Ethics Committees (MRECs) in the Netherlands (box 1) immediately recognized the need for accelerated review of COVID-19 research, which led to implementation of so-called 'fast-track-review-procedures' (FTRPs) to enable a swift start of urgent and relevant research. However, it is unknown how timelines in these FTRPs exactly differ from those of the regular review procedure (RRP). Also, it is unclear if quality of the review process differs between the FTRP and RRP and if certain review aspects were weighted differently.

Literature on previous evaluations among MRECs about ethical review of research during epidemics is relatively scarce. The Ethics Committee (EC) of the Henan Provincial People's Hospital published an overview of the review times of COVID-19-related research and issues encountered during the assessment [4]. Review processes and timelines of the Médecines Sans Frontieres (MSF) ethics review board (ERB) during the Ebola Virus Disease (EVD) epidemic were described by *Schopper et al* [5]. *Alirol et al* described the review of the WHO ethics review committee (ERC) during the EVD epidemic [6]. All reported expedited review times of on average a few days from submission to initial decision.

---

### Box 1. Organisation and regulation of medical ethical review in the Netherlands

In addition to the Central Committee on Research Involving Human Subjects (CCMO) (supervisor, ethics committee, and competent authority), the Netherlands has 17 accredited Medical Research Ethics Committees (MRECs) that review medical research with human subjects [2]. Each MREC consists of legally mandatory disciplines (physician, paediatrician, legal expert, methodologist, ethicist, lay member, medical devices expert, clinical pharmacologist, pharmacist) supplemented by other members. Pursuant to the Dutch General Administrative Law Act, a so-called reasonable assessment period of 8 weeks (56 days) normally applies to medical and scientific research subject to the Dutch Medical Research involving Human Subjects Act (WMO). This assessment period can be extended with a maximum duration of another 8 weeks. The WMO prescribes a specific decision period of 60 days which applies to research with medicinal products [3]. Researchers are given the opportunity to answer MREC comments in several comment rounds until approval can be given.

Since the beginning of the COVID-19 pandemic, many MRECs set up a FTRP which differed in content per MREC. No nationwide FTRP has been set up. However, the overall aim of all FTRPs was to give priority to COVID-19 research over regular research.

Review of studies on unregistered vaccines is reserved for the CCMO.

---

The need for ethics committees' preparedness during outbreaks or epidemics was already emphasized earlier. *Saxena et al.* presented six recommendations resulting from a World Health Organization (WHO) workshop with representatives of 29 countries, aimed to identify practical processes and procedures to facilitate ethics review during an infectious disease outbreak [7]. They recommended, for example, that "a national standard operating procedure (SOP) for emergency response ethical review should be developed and adopted by N(R)ECs and/or in-country competent authority" [7].

The objectives of this national evaluation of the FTRPs were:

1. To gain insight into the FTRPs of Dutch MRECs during the COVID-19 pandemic

2. Comparing the FTRPs with the RRPs from the perspectives of MRECs and investigators

Both objectives aim to learn from this COVID-19-period to be prepared for another pandemic and to improve review of medical scientific research in general. It is not unthinkable that FTRPs will once again be needed in the future.

## Methods

### Design

An explanatory sequential mixed method study was conducted. First quantitative data regarding timelines, review quality and satisfaction was collected using online questionnaires. These questionnaires were primarily developed on the basis of the experience with procedures of one MREC (METc VUmc). Questionnaires were used to quantify the FTRP nationally and to identify topics that could form a basis for a more in-depth evaluation conform the principles of Inductive Content Analysis [8]. Subsequently, as an explanatory follow-up we conducted group interviews resulting in qualitative data, which shed more light on the quantitative results [9].

### Data collection

In the first quantitative phase, in May 2020, a link to an online questionnaire containing questions about the FTRP, timelines, experiences and satisfaction was sent by email to all 20 chairs and 18 secretariats of all MRECs in the Netherlands. A separate email was sent including a detailed information letter about the study inviting both chairs and secretaries to complete the questionnaire. Another questionnaire and information letter were sent to 36 investigators of COVID-19 related research registered with the CCMO on 25th of April 2020. Email addresses were provided by the CCMO after written permission to share these data. It took about 15 minutes to complete the questionnaires.

In addition to the questionnaire data, review timelines of these COVID-19 related studies as well as comparable regular studies from the same time frame in 2019 were requested from the national registration system of the CCMO [10].

All invitees to the questionnaires were also asked to participate in a group interview. In the second qualitative phase, all participants who gave permission were invited to participate in a group interview. Four semi-structured online group interviews were conducted in June and July 2020 using Microsoft Teams: two group interviews with MREC representatives and two group interviews with COVID-19 investigators, in which saturation was reached. Characteristics of these group interviews are shown in Table 1.

Topic lists (Tables 2 and 3) with open-ended questions were developed based on questionnaire results.

**Table 1. Characteristics of group interviews.**

|  | Group 1 | Group 2 | Group 3 | Group 4 |
|---|---|---|---|---|
| Date | 22-6-2020 | 23-6-2020 | 30-6-2020 | 3-7-2020 |
| Participants | 4 MREC review officers | 4 COVID-19 investigators | 4 MREC review officers and 2 MREC chairs | 5 COVID-19 investigators |
| Interviewer | MJPAJ, PhD, male, ethicist MREC, assistant professor of medical ethics, supervised many interview studies | JvdP, MD PhD, male, chair MREC, obstetrician, trained to interview and teach residents and specialists | MJPAJ, PhD, male, ethicist MREC, assistant professor of medical ethics, supervised many interview studies | JvdP, MD PhD, male, chair MREC, obstetrician, trained to interview and teach residents and specialists |
| Duration | 1 hour 4 min | 52 min | 46 min | 54 min |

Interviews were led by one of the researchers (Table 1) with other research team members present. Interviews were Dutch-spoken, recorded using an audio-recorder, and transcribed verbatim.

## Data analysis

Descriptive analyses of the quantitative data were performed using IBM SPSS Statistics 26. A SPSS outlier analysis was applied and unexplainable negative timelines were excluded when

**Table 2. Topic list interviews MRECs.**

|  | Topic | Question |
|---|---|---|
| **Implementation process** | Method | How was the FTRP implemented and when? |
|  | Committee | Who exactly reviewed the COVID-19 submissions? |
|  | Documents | Which documents were required for review? |
|  | Central direction | Did you work together with a central coordinating committee? |
|  | Overview | Did you have a good overview of comparable studies? |
|  | Administrative aspects | Were administrative aspects handled differently? |
| **Review timelines** | Speed | How much faster was the review? |
|  | Influencing factors | Which factors expedited the review? |
| **Review experiences** | Submission quality | Did you experience differences in quality of the submissions? |
|  | Review quality | Did you experience differences in quality of the review? |
|  | Review aspects | Were some aspects weighted differently? |
|  | Strengths | What did you experience as the strengths of the FTRP? |
|  | Weaknesses | What did you experience as weaknesses of the FTRP? |
|  | Lessons learned | Which lessons can be learned to improve future review? |

**Table 3. Topic list interviews investigators.**

|  | Topic | Question |
|---|---|---|
| **Implementation process** | Prior to submission | Did you consult the MREC prior to submission about the procedure? |
|  | Publicity procedure | Did you know about a FTRP and if so, was it clear to you? |
|  | Documents | Which documents were required for review? |
|  | Submission quality | How was the quality of your submission and research proposal compared to regular research? |
|  | Administrative aspects | Were administrative aspects handled differently? |
| **Review timelines** | Speed | What is your opinion about the length of the review timelines and were these different compared to regular review? |
| **Review experiences** | Review quality | Did you experience differences in (quality of) the review? |
|  | Satisfaction | How satisfied were you with the review compared to regular review? |
|  | Points of improvement | How can the procedure be improved? |

analysing review timelines. An initial code structure based on the topic lists was developed to analyse the qualitative data. After review of the transcripts, the code structure was discussed during research team meetings and adjusted by modifying (sub)codes. Two researchers independently coded all four transcripts using the qualitative research program MAXQDA version 2018.2. Interpretation of codes was discussed until agreement was reached. Decisions made in the discussion sessions were debated during regular research team meetings. All researchers agreed with the final code structure, the encoded transcripts and the themes identified with help of the program MAXQDA [9].

### Ethics

This study was approved by the CCMO on 23rd April 2020. All invitees were provided with written information about the study and they were given the opportunity to ask questions. Participants gave written digital informed consent prior to completing the questionnaires, where they could also indicate whether they could be approached to participate in a group interview. In addition, group interview participants gave verbal informed consent prior to the interview. No compensation was provided to participants.

## Results

### Quantitative results

Twenty three (61%) out of 38 questionnaires were fully completed and returned from MREC representatives, representing 17 of the 18 MRECs in the Netherlands (94%). Twenty one of those 23 MREC respondents (91%), representing 16 of the 17 responding MRECs, indicated their MREC implemented a FTRP.

Furthermore, 20/36 investigators responded (56%). Sixteen (80%) of those respondents indicate they were experienced with submitting to a MREC in the past five years.

**Implementation process.** Seven out of the 16 MRECs established an 'ad-hoc subcommittee' to review the COVID-19 submissions. Other formations that reviewed these submissions were, for example, the full regular committee (ten MRECs) or the daily board (three MRECs). Nineteen (90%) out of the 21 MREC representatives indicated that it was not difficult (at all) to involve committee members in the FTRP. Eighteen (86%) out of the 21 MREC representatives indicated that their MREC required submission of the same documents as usual. However, 17/21 (81%) MREC representatives indicated that leniency was applied regarding the completeness of the initial submission. Nine (45%) out of the 20 investigators, however, did not notice any difference in leniency of the required documents compared to the RRP and 2/21 (10%) indicated even less leniency.

Eighteen (86%) out of the 21 MREC representatives indicated that no external consultation was needed for review.

**Review timelines.** Table 4 shows that the median total number of days from complete submission until MREC judgment was 10.5 days for COVID-19 research. Thereof, the median number of days attributable to the MRECs was 8.0 days. In comparison, for regular research, the median total number of days from complete submission to MREC approval was 98.0 days with a median number of days attributable to the MRECs of 50 days.

Sixteen (80%) out of the 20 investigators indicated that the review of their research was faster compared to regular submissions.

**Review experiences.** Eighteen (90%) out of the 20 investigators were (very) satisfied about the review process of their COVID-19 research, as were 19/21 (90%) MREC representatives.

Seventeen (85%) out of the 20 investigators felt their MREC took the urgency of the research into account and 16/21 (76%) MREC representatives indicated that the urgency of

**Table 4. Review times for COVID-19 research and regular research (based on data from national registration system CCMO).**

| Submissions | Total days from complete submission until MREC judgment Median (range) | MREC review days from complete submission until MREC judgment Median (range) |
|---|---|---|
| COVID-19 research (N = 28) *judgment between 13-3-2020 and 20-8-2020 and registered at 25-4-2020 (2 negative decisions)* | 10.5 (1–70) | 8.0 (1–56) |
| Regular research (N = 443) *judgment between 13-3-2019 and 20-8-2019 (0 negative decisions)* | 98.0 (1–363) | 50.0 (1–158) |

the research has weighted heavily in their review process. Nineteen (90%) out of the 21 MREC representatives indicated the quality of their review was equal to regular review, but only 13/21 (62%) indicated their review was 'as strict as in the regular situation', 4/21 (19%) indicated the MREC reviewed 'less strict' and 1/21 (5%) 'stricter'. Table 5 shows the number of respondents that indicated specific aspects were weighted differently in the FTRP.

There was no consensus among MREC representatives about the quality of the submissions in the FTRP: 9/21 (43%) indicated the quality was 'equal', 7/21 (33%) 'worse', 1/21 (5%) 'better' and the remainder had no opinion.

Five (24%) out of the 21 MREC respondents suggested the review of not COVID-19-related submissions was delayed due to the priority given to COVID-19 research.

## Qualitative results

In total, four semi-structured group interviews were conducted with 10 MRECs representatives and 9 investigators. Interviews ranged in length from 46 minutes to 1 hour and 4 minutes. Themes identified were structured on the basis of the three main aspects addressed in the interviews: 1. Implementation process 2. Review timelines and 3. Review experiences. Table 6 demonstrates the themes identified including how frequently they emerged per group.

**1. Implementation process.** *Review committee.* In general, COVID-19 review (sub)committees were composed relatively easy due to a sense of urgency resulting in (sub)committees that were ad hoc available.

"*We decided to set up a sub-committee of the large committee with at least all compulsory disciplines. . . And we always met when one or two protocols came in. . . then an ad hoc meeting was scheduled.*"

*(MREC representative 8)*

**Table 5. No. of respondents that indicated specific aspects were weighted differently in the FTRP compared to the RRP.**

| Review aspect | No. of MREC respondents (%) | No. of investigators (%) |
|---|---|---|
| Burdening of subjects | 0 (0%) | 1 (5%) |
| Legal aspects | 2 (10%) | 2 (10%) |
| Privacy aspects | 1 (5%) | 0 (0%) |
| Methodological aspects | 2 (10%) | 4 (20%) |
| Ethical aspects | 1 (5%) | 0 (0%) |
| Subject information | 1 (5%) | 1 (5%) |
| Administrative aspects | 8 (38%) | 6 (30%) |

**Table 6. Code structure and theme frequencies.** (one code per discussion point per individual).

| Structure | Theme | Number of passages investigators | Number of passages MRECs | Total number of passages |
|---|---|---|---|---|
| 1. Implementation process | Review committee | 3 | 7 | 10 |
| | Remote working | 0 | 9 | 9 |
| | Leniency | 23 | 9 | 32 |
| 2. Review timelines | Stakeholders involved | 6 | 0 | 6 |
| | Consequences regular research | 1 | 6 | 7 |
| | Priority | 4 | 7 | 11 |
| | Digitally working | 2 | 3 | 5 |
| | Urgency | 16 | 2 | 18 |
| 3. Review experiences | Quality of submission | 13 | 6 | 19 |
| | Quality of review | 12 | 19 | 31 |
| | Collaboration and coordination | 2 | 28 | 30 |
| | Communication | 29 | 5 | 34 |
| | Workload | 2 | 14 | 16 |

*Remote working.* Working remotely from home and using video conferencing probably contributed to rapid scheduling of review meetings.

"...what I think also helped is that everyone is sitting at home, so the external appointments were not there for the people so they were readily available every time, especially at the weekend."

(MREC representative 1)

"... that actually went very smoothly. Because everyone was sitting at home anyway so that, yes, I thought that it went surprisingly smoothly."

(MREC representative 5)

*Leniency.* Furthermore, although MRECs largely agreed that the submission had to be complete before approval of a study, leniency concerning the completeness of initial submissions was applied to accelerate the review process.

"*We were also satisfied with a simplified file at the start. So not the complete file, but protocol, IB, IMPD, that had to be there at the start in any case, and the other documents could come in phases*".

*(MREC* representative *4)*

"*All the signatures and such, we're not going to wait for that. As long as the essential documents are there*"

*(MREC* representative *1)*

"*We just thought it should all be complete. But let's say sometimes you had something that was less relevant to the substantive assessment that you said just if it will be there by the end of the week. In any case, it had to be there before we said our final 'yes'.*"

*(MREC* representative *6)*

**2. Review timelines.** Although investigators were generally pleased with the review time-lines and prioritization of COVID-19 research, they indicated that there was still room for improvement of timelines.

*Stakeholders involved.* Suggestions mentioned were, for example, giving preliminary feed-back shortly after submission and improving cooperation with other stakeholders involved, for example legal departments, medical technology departments and (MRECs of) other partici-pating centres.

> "*I think they still worked 90% harder and faster than they were used to. Only the 10% that remained was not yet in the time frame we thought of. We thought in hours and days, but 10% of what they are used to is not yet in hours and days. So that can result in some friction.*"

> *(Investigator 6)*

*Priority and consequences regular research.* MREC representatives also mentioned concerns about delayed appraisal of other research projects by giving full priority to COVID-19 research.

> "*Yes, that was also the case with us. It is mainly a case of trying to organize the entire process as efficiently as possible and getting the members in time . . .to vote for such an extra meet-ing. . .. . .. and to ensure that they have the documents and, as secretary, give priority to the assessment. And that doesn't detract from the diligence it's only . . . it just gets absolute prior-ity over other things. I think that's what the fast-track procedure envisaged.*"

> *(MREC representative 9)*

> "*As soon as there was covid on it, everyone suddenly went into top gear. Yes. . . and then I think*: *what about other research*? *Isn't that important*?"

> *(MREC representative 6)*

In particular, the processing of amendments to regular research was mentioned to be signif-icantly delayed. However, because much regular research was put on hold due to measures taken because of the pandemic, the consequences of this delay were minor.

> "*The regular protocols were indeed a bit less and you could easily fit them in and then give covid priority, but the amendments were and are a bit of a problem.*"

> *(MREC representative 9)*

*Digitally working.* In addition, the obligation to submit a wet signature cover letter for pri-mary submissions was suspended temporarily by the Competent Authority, which contributed to faster timelines.

> "*Now, of course, we were allowed to deliver some things just digitally and we were allowed to have digital signatures or a confirmation email. . . well, I think that has been very good for the environment because it saves a huge load of paper printing and scanning . . .and yes, why can't they keep it that way*?"

> *(Investigator 1)*

*Urgency.* Investigators indicated that fast timelines were crucial to be able to include enough patients, because the patient numbers were expected to decline in time. This sense of urgency

contributed to more dedication by investigators which also contributed to a faster review process compared to the non-pandemic situation. Investigators regretted, however, that they felt not all stakeholders involved in the research process were equally aware of this urgency or had other priorities related to the pandemic. Delays were particularly noted among legal, medical technology departments and participating centers.

"*Everyone is stressed out because it has to start*"

*(Investigator 1)*

"*Yes, if we had started a month earlier we would have finished the study and now eh. . .we are including 1 patient every 2 weeks. . .No eh. . .every week would have been a big gain.*"

*(Investigator 2)*

"*The momentum of the inclusion of participants was also very crucial, of course, because things are now going very differently. So we were really under enormous time pressure.*"

*(investigator 1)*

"*Yes, every day counted with the peak in patient numbers so it was also very urgent for our research.*"

*(Investigator 7)*

**3. Review experiences.** *Quality of submission.* Investigator respondents were unanimously satisfied with the quality of their submission which was said to marginally differ in quality compared to regular research. Some even indicated an improvement compared to regular research because they were forced to keep the research protocol simple and focussed due to time pressure.

"*If you also look at the quality. . .that is still just. . .I think we are still running a fine protocol with the same quality as say in a normal procedure so it seems like the procedure can just be done in this way without losing too much quality.*"

*(Investigator 2)*

"*In hindsight, I experienced this almost as a kind of blessing in disguise because in the past, we have sometimes had the tendency to set up studies in an enormous way and to make them complicated, and now we were forced to keep it fairly simple. And that has been a great benefit to the success of the study. That we were not tempted to make it too complicated. Yes, I actually found that very pleasant.*"

*(Investigator 7)*

"*Well, I think that efficiency in writing down a protocol in a short and concise way is a big advantage. Both in writing and in assessment. And I think we all have the tendency to write a whole lot of story around it in order to be as complete as possible. But this really forces you to get to the heart of the matter and to separate the essence from all those other things that are perhaps not necessary.*"

*(Investigator 7)*

MREC representatives however, mentioned that some submissions would probably have improved with a few weeks more preparation time. Amendments to initial submissions were submitted quickly. This was partly caused by the fact that new knowledge about COVID-19 became available every day and because extra centres had to be included in order to achieve the required inclusion rate.

"In general there was no difference in quality, but sometimes I had the feeling that if they had had three more weeks, the protocol would have been a bit better. Some things were not fully thought through and you also notice that now that protocols are being withdrawn again and that people are coming up with new information. . .yes, you are getting more and more information and I think that was the hardest thing about the whole covid thing: there was so much new information coming in every day. . . I personally and also the committee struggled with that, what to do with all the new information that was being poured over you."

(MREC *representative 6*)

*Quality of review.* Investigators and MREC representatives agreed that the quality of the FTRPs was not different from RRPs. MREC respondents were positive about maintaining quality because carefulness should take precedence over speed, but investigators indicated a clearer distinction between main and side issues would have been desirable.

"*And if you don't do your assessment well . . . that comes back in a negative way. So no, we just do the assessment just as we would do it otherwise. Just a little faster now.*"

*(MREC representative 2)*

"*Speed should not come at the expense of quality of assessment. That was an important starting point.*"

*(MREC representative 4)*

"*We have indeed not made any concessions in the broad outlines, but from my own experience with the things I have done, I have been less strict than usual on the dt's and t's, that you think, this can be nicer, this can be better.*"

*(MREC representative 1)*

"*Well, they didn't spare us with the comments. They just gave us the comments more or less like we actually always expect, sometimes more than the other. When I look at the comments, I infer that the assessment was done as always with respect for the basic quality requirements that we set for each other.*"

*(Investigator 9)*

"*In the first assessment, I got 33 questions and they were a bit of the same nature as usual. And um. . . most of them were good content-related questions.*"

*(Investigator 3)*

*Collaboration and coordination.* Especially among MREC respondents a need for collaboration and coordination between research projects was experienced. Many MRECs took the

feasibility in terms of recruitment of patient numbers into account, but they indicated having not always an accurate overview of the availability of patients and competing interests of different research questions. MRECs were for example concerned about the considerable amount of human tissues that was collected per patient due to inclusion in multiple studies. On some locations this was solved by a central coordinating committee.

"*Well, at a certain point we also started to get worried, especially about the amount of blood that was being taken from people, but at that point the hospital actually took the initiative to create a central point where all the tests were registered and they looked at what could be carried out.*"

*(MREC representative 1)*

But there were also concerns about conducting research that was already carried out elsewhere. It was not easy for MRECs to gain insight into this. Particularly for intervention studies, the short peak of the pandemic made it difficult to include enough participants. More central management and collaboration between different research groups and MRECs could improve this.

"*I found a lot of overlap and too little coordination. And that is also a bit how the country works. Maybe we should do it differently next time. There was also the speed of, let's say, the epidemic, and before some protocols were started it had already disappeared. Yes, this is particularly true of intervention studies involving medicines. They all have great problems filling their N now. Then of course they hope for a second wave, I don't hope for that by the way, but it would have been prevented if they had joined forces I think.*"

*(MREC representative 6)*

Furthermore, the respondents also felt room for improvement was needed with regard to the cooperation and coordination between MRECs. There is a need for a standard procedure that works nationally. Now each MREC implemented its own FTRP.

"*Yes, I just think a very clear eh. . .. General policy. In which METCs hopefully also consult with each other that in such cases certain things should be uh..postponed, for example, that a slightly less extensive first protocol or so should be assessed and that later on you should see what else is necessary.*"

*(Investigator 4)*

"*Everyone is doing their own thing. A bit more centralised control could play a role in getting it even better. Certainly with medication studies.*"

*(MREC representative 6)*

*Communication*. In general, there was clear communication about when an application should be submitted and discussed in a MREC meeting. However, according to investigators, it was not clear when a response from the MREC could be expected.

"*It was very clear that there was an urgent procedure. And there were also deadlines on the website. . . only what I just indicated. . . that didn't quite match what was being discussed in the corridors. We were given the impression that. . . er. . .yes, the assessment would be quicker*

*than it ultimately turned out to be. So the deadline for submission was clearly stated, only the subsequent speed of the assessment was not clearly indicated. And if you can only submit everything in the mailbox, it remains unclear what the next plan of action will be with the documents you have submitted.*"

*(Investigator 2)*

Moreover, telephone reachability of the secretariats decreased due to working remotely and one could not physically encounter the MREC office due to the COVID-19 restrictions.

"*Well, accessibility of the secretariat by phone was not very good. It was. . . it was possible by email, but it was difficult by phone. Anyway, if you call the chairman, that makes a difference. But he will undoubtedly not always like it if I call him that way again in the next projects that are not covid related.*"

*(Investigator 6)*

This lack of clarity and communication options in an urgent situation led to frustration among investigators. A designated contact person, frequent MREC consultation and verbal clarification of comments improved communication and limited inconsistency in advice.

"*What I found really annoying is that I got a different contact person every time. So if I gave a follow-up, an answer to an email, I would get an answer from a totally different person and they would repeat things or ask for confirmation again and that I thought was a great pity.*"

*(Investigator 1)*

*Workload.* The FTRP meant a significant change to the working processes at the MREC secretariats. Daily routine changed dramatically overnight and workload increased. Besides the great flexibility that was required of both MREC members as secretaries, there was also social and scientific pressure to process everything very quickly. The small decline in regular work because of temporarily halted research projects helped to make it manageable. However, logistics, all remote, were a burden on the secretariats.

"*Precisely those little things, all together—a phone call here, a phone call there—seeing that you have the right people together at the right time, drafting emails, that just takes a lot of time for the secretariat.*"

*(MREC representative 3)*

## Discussion

This study showed that accelerated review is feasible during emergency situations, without compromise on perceived review quality and to the satisfaction of both investigators and MRECs. Review timelines of MRECs were more than four times as short in FTRPs compared to RRPs. This shortening may be explained by installing (sub)committees that were prepared to meet outside office hours, the sense of urgency of those involved, full priority given to COVID-19 research, regular research put on hold due to COVID-19 measures, the use of online review meetings and administrative leniency, for example with regard to (wet) signatures. The data found indicate that, according to the respondents, leniency regarding administrative matters do not in any way affect the ethical rigour of the review process. Ethical

principles that apply to medical scientific research in general equally apply to COVID-19 studies. This finding is in agreement with the position paper of the European network of Research Ethics Committees (EUREC) that concludes: *". . . the pressure currently being exerted on medical research must not lead to research or testing of pharmaceuticals on humans without complying with the ethical standards applicable to medical research."* [11] Thus, the sense of urgency of COVID-19 research that is widely acknowledged, seems not to affect accepted ethical and legal standards.

In addition, the FTRP is associated with a high degree of satisfaction among both investigators and MRECs.

Weaknesses of the FTRP experienced by both investigators as MRECs were a lack of overview of ongoing COVID-19 studies and insufficient collaboration between investigators as well as the different MRECs. This complicates an expedited review. Therefore, closer collaboration between investigators, institutions as well as MRECs is needed. This is endorsed by conclusions from previous papers [5–7, 12, 13]. Local coordinating committees could prevent too much (low-quality) research being conducted in one specific group. In addition and as a result of prioritizing COVID-19 research applications, delayed review of (amendments to) regular research can occur. These capacity issues should be taken into account, as was also suggested by Ma et al. [13].

The limited period of time during which sufficient patients were available for inclusion made an expedited review all the more important. Short communication lines, a designated contact person, being easily reachable, focus on essentials and increasing the sense of urgency in all involved in conducting the research were identified as factors which can contribute to an even faster review process and thus to a higher chance of reliable study outcomes.

From the challenges and problems that were perceived by the respondents several lessons can be learnt. First, it has become clear that in emergency situations review timelines can be significantly reduced, without compromising on ethical and legal principles and guidelines. It is not unfeasible that in the future also RRPs can be handled in similar timelines. This may imply that also in RRPs administrative aspects (e.g. the wet signature or incomplete submissions) will be reviewed with some leniency. While main issues regarding ethics and law need to be reviewed with rigour, leniency regarding side issues may further speed up the review process and thus stimulate scientific progress. Working remote has also facilitated the organisation of MREC meetings very shortly after submission. Online meetings may also in the future speed up the review process. In this respect, the perceived challenge is that working from home should not in any way negatively affect the reachability of MREC secretariats. Another lesson that was learned relates to the importance of coordination and cooperation of MRECs on the national and international level. National and international overview of studies that are conducted in human subjects can prevent unnecessary overlap in research. Importantly, this may also prevent unnecessarily exposing human subjects to risks and burdens of participation in studies that are also conducted elsewhere.

In conclusion, our study provides a comprehensive overview of the fast-track review of COVID-19 research in the Netherlands during the COVID-19 pandemic. The study shows the importance and the possibilities of much faster review during emergency situations without perceived quality loss. This all touches upon the heart of research ethics principles. More research is needed to find out how lessons learned can best be implemented so that a standard (national) review procedure can be developed to be prepared for future emergency situations. Furthermore, since the way in which medical ethical review is organized and regulated by law varies internationally, we also encourage more (expedited) review procedures to be evaluated and shared internationally. This can be a first step towards more international collaboration prior to and during an epidemic.

## Limitations

This study was limited by the data available and the response rates. The administrative data used to calculate review timelines concerned data entered to the CCMO database by MRECs themselves. There is no control over this data. Furthermore, the number of participants in the group interviews was limited. However, since saturation has been achieved we do not think this affected the generalizability of the results of our study. All MRECs in the Netherlands were invited to participate, including the MREC to which we are affiliated ourselves. As this involved only one of the MREC participants and one of the investigators and openness was encouraged during the interviews, we think this did not affect the results. Lastly, some interesting topics, such as patient perspective and implications of delaying appraisals or regular research fall outside the scope of this study due to ethical or privacy restrictions or limited availability of data. Further research is needed to investigate this.

## Supporting information

**S1 File. Questionnaire (principal) investigators (English).**
(PDF)

**S2 File. Questionnaire MRECs (English).**
(PDF)

**S3 File. Questionnaire (principal) investigators (Dutch).**
(PDF)

**S4 File. Questionnaire MRECs (Dutch).**
(PDF)

**S5 File. Topic list interview investigators (Dutch and English).**
(PDF)

**S6 File. Topic list interview MRECs (Dutch and English).**
(PDF)

**S7 File. COREQ checklist.**
(PDF)

**S8 File. Dutch quotes interviews.**
(PDF)

## Acknowledgments

The authors thank the CCMO for making data available for this research. They also thank Peter van de Ven, methodologist and Caroline Terwee, methodologist, for commenting on the design of the study and Michel Paardekooper, privacy officer, for his advice on data protection. Finally they thank Richard Dekhuijzen, chair MREC CMO Arnhem-Nijmegen and chair NVMETC, and Louis Bont, chair MREC Utrecht, and Jan Swinkels, chair MREC Academic Medical Center Amsterdam, for commenting on a draft of this paper and Madeleine Evers for checking the manuscript for correct English.

## Author Contributions

**Conceptualization:** R. IJkema, M. J. P. A. Janssens, J. A. M. van der Post, C. M. Licht.

**Data curation:** R. IJkema.

**Formal analysis:** R. IJkema, C. M. Licht.

**Investigation:** R. IJkema, M. J. P. A. Janssens, J. A. M. van der Post, C. M. Licht.

**Methodology:** C. M. Licht.

**Project administration:** R. IJkema.

**Supervision:** J. A. M. van der Post, C. M. Licht.

**Writing – original draft:** R. IJkema.

**Writing – review & editing:** M. J. P. A. Janssens, J. A. M. van der Post, C. M. Licht.

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
