## [Decision Letter · Decision Letter 0]

8 Apr 2021

PONE-D-21-05904

Medical ethical review of COVID-19 research in the Netherlands; a mixed-method evaluation among Medical Research Ethics Committees and investigators

PLOS ONE

Dear Dr. IJkema,

Thank you for submitting your manuscript to PLOS ONE. After careful consideration, we feel that it has merit but does not fully meet PLOS ONE’s publication criteria as it currently stands. Therefore, we invite you to submit a revised version of the manuscript that addresses the points raised during the review process.

We look forward to receiving your revised manuscript.

Kind regards,

Prof. Ritesh G. Menezes, M.B.B.S., M.D., Diplomate N.B.

Academic Editor

PLOS ONE

Journal Requirements:

Please include additional information regarding the survey or questionnaire used in the study and ensure that you have provided sufficient details that others could replicate the analyses. For instance, if you developed a questionnaire as part of this study and it is not under a copyright more restrictive than CC-BY, please include a copy, in both the original language and English, as Supporting Information.

Furthermore, please provide additional information regarding the questionnaire development process, including the theories or frameworks which were employed.

When reporting the results of qualitative research, we suggest consulting the COREQ guidelines: http://intqhc.oxfordjournals.org/content/19/6/349. In this case, please consider including more information on the number of interviewers, their training and characteristics; and please provide the interview guide used.

Please provide additional details regarding participant consent. In the ethics statement in the Methods and online submission information, please ensure that you have specified (1) whether consent was suitably informed and (2) what type you obtained (for instance, written or verbal). If your study included minors under age 18, state whether you obtained consent from parents or guardians. If the need for consent was waived by the ethics committee, please include this information.

We note that you have indicated that data from this study are available upon request. PLOS only allows data to be available upon request if there are legal or ethical restrictions on sharing data publicly. For information on unacceptable data access restrictions, please see http://journals.plos.org/plosone/s/data-availability#loc-unacceptable-data-access-restrictions.

3a) If there are ethical or legal restrictions on sharing a de-identified data set, please explain them in detail (e.g., data contain potentially identifying or sensitive patient information) and who has imposed them (e.g., an ethics committee). Please also provide contact information for a data access committee, ethics committee, or other institutional body to which data requests may be sent.

3b) If there are no restrictions, please upload the minimal anonymized data set necessary to replicate your study findings as either Supporting Information files or to a stable, public repository and provide us with the relevant URLs, DOIs, or accession numbers. Please see http://www.bmj.com/content/340/bmj.c181.long for guidelines on how to de-identify and prepare clinical data for publication. For a list of acceptable repositories, please see http://journals.plos.org/plosone/s/data-availability#loc-recommended-repositories.

Reviewers' comments:

Reviewer's Responses to Questions

**Comments to the Author**

1. Is the manuscript technically sound, and do the data support the conclusions?

Reviewer #1: Partly

Reviewer #2: Yes

Reviewer #3: Yes

Reviewer #4: Partly

Reviewer #5: Yes

Reviewer #6: No

Reviewer #7: Yes

Reviewer #8: Partly

2. Has the statistical analysis been performed appropriately and rigorously? 

Reviewer #1: I Don't Know

Reviewer #2: Yes

Reviewer #3: I Don't Know

Reviewer #4: No

Reviewer #5: I Don't Know

Reviewer #6: No

Reviewer #7: I Don't Know

Reviewer #8: N/A

3. Have the authors made all data underlying the findings in their manuscript fully available?

Reviewer #1: Yes

Reviewer #2: Yes

Reviewer #3: Yes

Reviewer #4: No

Reviewer #5: No

Reviewer #6: Yes

Reviewer #7: No

Reviewer #8: No

4. Is the manuscript presented in an intelligible fashion and written in standard English?

Reviewer #1: Yes

Reviewer #2: Yes

Reviewer #3: Yes

Reviewer #4: No

Reviewer #5: Yes

Reviewer #6: No

Reviewer #7: Yes

Reviewer #8: Yes

5. Review Comments to the Author

Reviewer #1: 1. Box 1 states, "a so-called reasonable assessment period of 8 weeks (56 days) normally applies to medical and scientific research subject...." Comparing this to your data in Table 1, the median review days for regular research: 50.0 (1-158) shows that the 56 days spec can be far exceeded. What is the cause?

2. I note in Table 1 that basically everything get approved! Out of 471 submissions, only 2 were rejected. That is a reject rate of 0.4% -- this is very, very low. Someone can interpret this as a low quality standard -- or, are you sending these submissions back and forth over and over to the PI for corrections, until you give an approval?

3. What the revise and resubmit less or more with the fast-track process? How many cycles in each (regular and fast track)?

4. What about the issue of submissions returned to the PI as they were out of scope (not WMO)? This is known to be a problem in the Netherlands because it is very difficult to determine what is in scope for WMO as the definition is vague and there is the ambiguity of biopsych/psych research. This would apply to COVID as well due to the mental health and isolation issues associated with the topic. Did MRECs find more of this happening with the COVID submissions?

5. Line 170, are you saying that a digital signature is not permitted (even before COVID)? This seems unusual and burdensome.

6. There is no real explanation of what the fast track process is -- does it mean that it gave a priority to COVID research (over regular research?) or was it that there was an additional reviewing mechanism (a funnel) that took those submissions while there was no change to how regular research submissions were managed?

7. I don't see a formal measure of review quality. How was that actually measured?

8. What about the issue of the capability of the MREC members to review COVID research? Did you need external experts? How did you find them? Did this impact review time (looking for them)? Or did you have the on-site/internal COVID expertise to review those submissions? This is important to consider in light of the problem with HRECs approving research that is low quality/poor design . See and perhaps reference this paper, https://jme.bmj.com/content/46/12/803

Reviewer #2: This is a timely and well-written paper on a relevant topic (ethics reviews of COVID research). I have some minor suggestions for improvement:

Introduction: A bit more information is required about the functioning of the MREC's in The Netherlands (eg what are typical member's profiles), so that readers can compare these to the working of the MREC's in their own country.

p6 "As an explanatory follow-up the quantitative results were further explained with in-depth qualitative data"

 'explained' is not really an appropriate term as qualitative and quantitative research have different finality, I would suggest changing it to 'shed more light on'.

p6: how many people were there in each focus group? Best to name exact numbers per group also here. It would be good to provide a table with demographics of focus group participants as well.

Am I correct that the factual data such as the review times were asked in the questionnaire? So these were based on self-reporting of the participants rather than on actual administrative data? Best to make this clear in the text.

p10: "Working remotely from home and using video conferences probably contributed to a rapid scheduling of review meetings."  is this what the focus group participants said or is it your own interpretation? Make sure to distinguish this in the results of the qualitative research.

p13: "The little decline"  the small decline

The discussion of the qualitative research is very concise. I feel that this would be helped if some quotes from the transcripts were provided to illustrate the themes.

At the end of the paper, there are recommendations in a table, which is fine. However, I would expect a bit more elaboration of these recommendations:  what would these practically entail and how do they relate to your research findings? There is already something in the text preceding the table, but I feel this needs a bit more structure, and the recommendations themselves have to be made more visible in the text.

Reviewer #3: This paper presents research focused on the experience of researchers and MRECs during the beginning of the COVID-19 pandemic in 2020. The researchers used both administrative data on review length and novel data (both quantitative and qualitative) focused on the experience of reviewers and submitters to understand how the fast track review process (FTRP) impacted COVID-19 research. Overall, the paper does a nice job showing the need for the FTRP, as well as the experience on both sides. The paper should include more technical information, such as more detail about the questionnaire process, any relevant demographics for respondents, how data was analyzed. There are some interesting points in the data that are not reflected in the writing.

Major issues:

1. Please provide more detail about the questionnaire and the semi-structured interview guide; it’s very unclear how they were designed, what was asked, how long they took, what the hypotheses were, etc.

2. Some of the more interesting findings are not focused on as much – investigators unanimously thought their research was either equal or better than usual, while reviewers that it was equal or worse (33%!); investigators also seemed to express frustration at legal, tech, participating centers for not moving fast enough. This pokes at a really interesting ethical issue – you have researchers not spending enough time on their own research plans, believing they are as good or better than their other research, and feeling frustrated that legal is concerned…combined with reviewers feeling pressured to approve quickly. All of the recommendations are focused on getting approvals done even more quickly, with nothing focused on the ethics of rushing approvals for protocols that are not adequate.

3. A few references to patients and patient burden – should make clear why patients were not contacted for the study; it would be interesting to at least comment on patient experience here, or discuss as a limitation

Minor issues:

1. 115-118: Provide response rates for individuals, not by MRECs. Provide details on recruitment, follow-ups, compensation

2. In Implementation Process – it’s confusing to jump between MRECs as entities and individual respondents as entities

3. In total, four semi-structured group interviews were conducted with 10 MRECs representatives and 9 investigators – mixed or by group?

4. A few spelling and grammar errors throughout – nothing major, just recommend a close read to make sure all errors are caught

Reviewer #4: Summary of the manuscript: This paper aims to evaluate the fast-track review procedures (FTRP) set up by medical research ethics committees (MREC) in the Netherlands during the COVID-19 pandemic in an effort to meet the urgent need for accelerated review of research proposals. It is an exploratory sequential mixed-method study that uses online questionnaires and in-depth interviews. The authors found that the total number of review days was shorter in FTRP than in regular review procedures (RRP) but this did not seem to impact the quality of review. The main difficulties of the FTRP were the heavy workload for MREC members, the lack of accessibility/coordination between stakeholders, and the lack of “overview of COVID-19 research”. The authors end the paper with a table of recommendations for more efficient and less burdensome FTRPs in the future. Furthermore, in the discussion, the aim is re-phrased so that the goal of the study is to identify differences between FRTP and RRP, rather than to evaluate FRTP in a medical ethical context. Overall, the paper succeeds in giving a description of the FRTP process by MRECs during the COVID-19 pandemic, but not an evaluation or analysis.

Strengths: This paper is ambitious and its objective is very important, as 2020 has been a unique opportunity for examining medical ethical review processes. It also appears methodologically sound. The study received ethical approval and participants provided informed consent. The results are quite striking and thought-provoking, and there is a good, broad summary of the qualitative results.

Weaknesses: The results are straightforward, but lacking in-depth analysis and ethical reflection. It is also unclear what questions were asked in the questionnaire and interviews. The language of the paper is overall moderately difficult to follow, so the paper would benefit from thorough copyediting.

Major issues:

1) It is unclear what questions were asked in the online questionnaire (Line 84-85) and how the interviews were structured (e.g. the topics that were planned to be discussed) and coded (line 99-105) This would be helpful if the authors could include a sample of the questionnaire and the coding, e.g. as part of an appendix/supplementary information.

2) The quantitative results section would benefit from a more rigorous statistical analysis, e.g. to identify patterns or trends, and/or differences between investigator findings and MREC findings. Currently, the results are stated without detailed explanation or comparison e.g. Line 143-155

3) The qualitative results section would benefit from more detailed description of the interview findings. For example, Line 169: explore the “leniency” further. Line 179-183: How much impact did the delay of regular research have on the investigators? It would further be important to analyse the ethical implications of delaying appraisals of regular research, perhaps in the discussion section.

4) The discussion section seems to re-state the results. It would benefit from an in-depth ethical analysis, e.g. the implications of “leniency” of required documents, or the increased need for human tissues (line 211-212.) The paper would also benefit from a discussion of the strengths/weaknesses of the study, e.g. sample size, subjective nature of the data (like results mentioned in Line 194-195), and applicability of the findings.

5) The discussion section would benefit from a description of the recommendations before outlining them in a table, as well as giving examples and evaluating the benefits, costs, feasibility, and justification of the recommendations.

6) While some of the findings are endorsed by previous papers (Line 260) the points in discussion section need to be tied back into the existing literature to place the findings in a wider context, especially when analysing the ethical implications or providing recommendations.

7) Overall, the language, sentence structure, and (lack of) punctuation decrease the readability of this manuscript. This also makes some of the results unclear. For example: line 210-211 - the protocols did not have a “good overview of the availability of patients and possibilities in the hospital”?

Minor issues:

8) The tables are not very easy to read; they could benefit from more formatting

Reviewer #5: This manuscripts reports the results of a mixed method student into the fast track reviewing procedures that medical research ethics committees (MRECs) in the Netherlands implemented during the COVID-19 pandemic in 2020. Overall, the research design is sound and the data reported supports the conclusions, and the results are of interest to an international audience as human research ethics committees worldwide have implemented similar fast track review procedures throughout the pandemic. However, several issues need attending to bring the manuscript up to a standard applicable for an international journal and audience. I have made many suggestions in the comments of the PDF manuscript file (attached). Notably, the authors should append into the Supp Materials a completed COREQ checklist for qualitative research as well as the the survey questions and interview guide. In the manuscript they should provide an assessment of response/compliance rates and address some discrepancies in responses shown in Table 2.

Substantively, while I concur with the authors recommendations for improving the fast track procedures of MRECs, it is unclear why they could also not be for regular review processes, as it seems that this system is in need of improvement as well – 90 days for regular review seems excessive and a national approach may help to reduce duplication. I’m unsure if an exceptional case can be made – why not just improve the whole review system while we are here?

Finally, the manuscript contains some awkward phrasing throughout and should be thoroughly edited for an English-language international journal.

Reviewer #6: Introduction

1. It is not clear why the information about the Dutch setting has been put in a box. I would suggest this material is simply described in the introduction section. And that Dutch names of legislation are excluded.

2. Although the literature of ethics review of research in pandemics may be limited, it would be helpful if the authors outline the key findings of this research and WHO recommendations in more detail in the introduction section, including the few publications regarding ethics review in the current pandemic.

3. The aims and importance of the study are not clearly stated at the end of the introduction section

Methods

4. The methods are currently described in insufficient detail. More details are needed regarding the development, contents, implementation, and analysis of the survey, and regarding the design, implementation, and analysis of the group interviews. It would be preferable if the authors followed reporting guidelines.

Results

5. Quantitative results section is difficult to follow and I would suggest that authors revise to improve clarity and readability. It would be helpful is the authors provide demoniators numerators, and a table provide full results.

6. The authors should consider adding a table that provides an overview of the interview codes and subcodes identified.

Discussion

7. The discussion section should begin with a clear statement of the key findings of the study. Currently, there is too much repetition of study aims and results.

8. The discussion of the results in the context of existing literature is not sufficient, partly reflected by the article citing a total of 11 publications. Although there may be limited reearch focusing on ethics review of research in pandemics, there is certainly relevant research that the authors should draw on.

9. The practical implications of the research need to be discussed in more detail. Recommendations are listed in table 3, but the authors should discuss these in the discussion section. As these recommendations are from the authors, the summary should also be in a box and not a table.

10. The limitations of the study are not described.

Reviewer #7: In this mixed-methods study, the authors evaluate how expedited research review in the Netherlands during the COVID-19 pandemic affected processes and perceptions form both investigators and board review members. This is one of the first studies of its kind to use mixed-methods techniques to study implementation of expedited research review during COVID, and as such is likely to be of significant interest to the research community. The generalizability of this study is questionable given that nation-wide efforts expedited review processes were rare in countries like the US, where research review is usually performed by single institutions. However, as these institutions likely had COVID fast track policies, there are still learning lessons.

A few comments:

INTRODUCTION

1) Some further detail on the FTRP in the introduction could be helpful. Did the FTRP apply only to COVID-related research? Did it apply only to prospective studies, or retrospective/observational studies as well? Furthermore, was other non-COVID research allowed to be submitted for review during the COVID period? Did these policies vary by institution or were they set country-wide?

METHODS

My biggest issues apply to the qualitative portion

2) The authors should follow a published checklist, such as the COREQ checklist, for reporting their qualitative study.

3) The authors should describe how saturation of interview themes during the interview period was arrived at.

4) In the methods, The number of interview guide questions should be included, even though the guide itself is provided as a supplement.

5) A weakness in this paper is the lack of a cited theoretical model or foundation for the qualitative interviews. How were codes arrived at for the qualitative portion? Was a grounded theory approach, or another method used?

6) To calculate review timeline, were institutions required to send all of their COVID reviews for analysis, or was there some sort of convenience sampling where institutions were allowed to choose which COVID reviews to send for analysis?

RESULTS

7) The manuscript would significantly benefit from inclusion of direct quotes from interview transcripts to support some of the general points made. Alternatively, a table of representative quotes within each theme would be beneficial.

8) How were the number of 4 total interview groups arrived at? How many participates participated in each group?

Reviewer #8: 1. I would first like to note the importance of this topic and the potential impact of this research. I especially respect the integration of qualitative data here, and I appreciate this research team’s identification and exploration of such an important issue.

2. I’m concerned that the level of detail provided in the Methods section is insufficient to understand the data; in particular, expanding the description of the instrumentation would be helpful to understand the meaning of the data; e.g., how were these surveys/interview guides developed? Was there cognitive interviewing or pilot testing? Were they based on any validated instruments, literature, etc.? What data was collected via survey versus secondary analysis of CCMO records, and how were these integrated?

3. Further, additional explanation and description of the coding and thematic analysis would be helpful to understand the level of rigor involved in the methodology; e.g., did the coders measure any sort of inter-coder agreement? I do see that the coders “independently” applied the codebook to the transcripts, but am not clear on whether they each coded all 4 transcripts in their entireties. And were these a priori structural themes? Emergent thematic?

4. At several points throughout, there was some confusion re: the extent to which statements were actual presentations of data, versus conjecture/observation/opinion of the authors; e.g., “More central management and collaboration between different research groups and MRECs could improve this.” [216-7] -- it is unclear if this is an actual finding emerging from the data or just an argument from the authors. There are several similar instances throughout.

5. The presentation of quantitative results as separate from qualitative results—and the descriptor of “mixed method study”—is confusing. According to the authors’ statement that “as an explanatory follow-up the quantitative results were further explained with in-depth qualitative data” [81-2], it seems more informative/appropriate to present these results in a cohesive, synthesized analysis. Some qualitative researchers would argue that collecting follow-up qual data from participants to further inform their prior quant responses does not constitute a “mixed methods” study design. I don’t have a strong opinion on this, but I wanted to point this out so the authors can revise to clarify or preemptively defend against potential critiques/confusion.

6. I’m somewhat concerned about the framing of investigators being “satisfied” with the review process (e.g., [143-4])—investigator satisfaction is not the goal of any ethics review in any circumstances, and should not be messaged as a measure of quality, success, or efficiency.

7. I’m also struck by the reporting of quant/qual data re: MERC reps’ quality assessment of reviews, only insofar as a MERC rep could reasonably have patent or latent bias in that assessment; i.e., if standard review is the minimum threshold for ethics review, why would a MERC rep ever consider their review to be below the ethical standard, much less discuss that openly with other MERC reps? For the sake of clarity: theoretically, the MERC reps who reviewed these studies would *by definition* believe that their review was appropriate, otherwise they would not have done it that way. Further, given the method chosen here (group interviews), a rep who questioned the quality of a review would have little-to-no incentive to disclose that doubt amongst their peers. But perhaps I’m missing something here.

8. Overall, I worry that the authors may have limited the value/applicability/impact of their study by presenting what appears to be a cursory analysis, with the results section comprising little more than a recitation of (presumably structural) themes. Additional themes, further exploration, or just a more extensive analysis would be helpful to better support the study’s overall impact.

9. The discussion of other review timelines seems out of place and uninformative considering the wide range of legal/regulatory requirements in oversight/review, additional systemic burdens from the pandemic, etc.

10. It would be helpful to provide some amount of qualitative data to make sense of the themes presented; without this, readers cannot know how salient a theme was, if 2 or more themes were from the same groups/participants, etc.

11. “These changes did not affect the quality of the review. Moreover, the FTRP is associated with a high degree of satisfaction among both investigators as well as MREC members.” [252-3] Is this first statement demonstrably true based on the data? Or is this an argument by the authors? There doesn’t seem to have been any actual measurement of quality beyond the perceptions of those who were (i) being reviewed or (ii) doing the very review of which the quality is in question.

12. The overall framing of the results and discussion is focused on investigator and MERC rep perspectives, with little-to-no acknowledgment of competing priorities (e.g., clinical care, allocation of limited resources); i.e., to recommend any action/policy without representation of additional stakeholders seems unwise at best.

13. Some attention is needed for minor grammatical and typographical errors.

6. PLOS authors have the option to publish the peer review history of their article (what does this mean?). If published, this will include your full peer review and any attached files.

Reviewer #1: No

Reviewer #2: No

Reviewer #3: No

Reviewer #4: No

Reviewer #5: No

Reviewer #6: No

Reviewer #7: No

Reviewer #8: No

---

## [Author Response · Author response to Decision Letter 0]

6 Jun 2021

We checked if our manuscript meets PLOS ONE’s style-requirements including file naming and adjusted where needed. 

Copies of the questionnaires are included as Supporting Information in both Dutch and English. The topic lists are included in the manuscript as table 2a and 2b. In addition they are included as Supporting Information in both Dutch and English.

De-identified data sets regarding the questionnaires were uploaded to the DANS data repository: https://doi.org/10.17026/dans-z2r-hnfa

Furthermore, please provide additional information regarding the questionnaire development process, including the theories or frameworks which were employed.

Since this topic concerns a unique emergency situation, questionnaires were primarily developed on basis of the experience with procedures of our own Ethical Review Committee (METc VUmc). Questionnaires were used to quantify the Fast Track Review Procedure nationally and to identify topics that could form a basis for a more in-depth evaluation conform the principles of Inductive Content Analysis (Kyngäs H. (2020) Inductive Content Analysis. In: Kyngäs H., Mikkonen K., Kääriäinen M. (eds) The Application of Content Analysis in Nursing Science Research. Springer, Cham. https://doi.org/10.1007/978-3-030-30199-6_2). 

Information was added to the manuscript (Design page 6.). 

When reporting the results of qualitative research, we suggest consulting the COREQ guidelines: http://intqhc.oxfordjournals.org/content/19/6/349. In this case, please consider including more information on the number of interviewers, their training and characteristics; and please provide the interview guide used.

The COREQ checklist was used and added as Supporting Information. More information on the interviewers, their training and characteristics was added to the manuscript (table 1). The interview guides were added as table 2a and 2b.

Please provide additional details regarding participant consent. In the ethics statement in the Methods and online submission information, please ensure that you have specified (1) whether consent was suitably informed and (2) what type you obtained (for instance, written or verbal). If your study included minors under age 18, state whether you obtained consent from parents or guardians. If the need for consent was waived by the ethics committee, please include this information.

Under ‘Ethics’ in the methods section information about (type of) informed consent was added (page 9).

3a) If there are ethical or legal restrictions on sharing a de-identified data set, please explain them in detail (e.g., data contain potentially identifying or sensitive patient information) and who has imposed them (e.g., an ethics committee). Please also provide contact information for a data access committee, ethics committee, or other institutional body to which data requests may be sent.

Our study made use of three different types of datasets:

a. Datasets questionnaires: a. MRECs b. investigators. 

These datasets were de-identified and uploaded to the DANS data repository: https://doi.org/10.17026/dans-z2r-hnfa

b. Datasets obtained from the national registration system of the CCMO. 

Publicly sharing these data is restricted by the CCMO. Requests can be sent to metc@vumc.nl. 

c. Interview transcripts and analysis files. Since it is not possible to fully anonymize these data because of potentially/indirectly identifying clauses, it is not ethical to share these publicly. Data requests can be sent to metc@vumc.nl. 

3b) If there are no restrictions, please upload the minimal anonymized data set necessary to replicate your study findings as either Supporting Information files or to a stable, public repository and provide us with the relevant URLs, DOIs, or accession numbers. Please see http://www.bmj.com/content/340/bmj.c181.long for guidelines on how to de-identify and prepare clinical data for publication. For a list of acceptable repositories, please see http://journals.plos.org/plosone/s/data-availability#loc-recommended-repositories.

The de-identified datasets were uploaded to the DANS data repository: https://doi.org/10.17026/dans-z2r-hnfa

Review Comments to the Author

Reviewer #1: 

1. Box 1 states, "a so-called reasonable assessment period of 8 weeks (56 days) normally applies to medical and scientific research subject...." Comparing this to your data in Table 1, the median review days for regular research: 50.0 (1-158) shows that the 56 days spec can be far exceeded. What is the cause?

Legally, the Ethics Committee has a maximum of 8 weeks (56 days) to reach a decision, unless the MREC of CCMO has given notice of requiring more time. The extension can also have a maximum duration of 8 weeks (56 days) (www.ccmo.nl). 

In our 2019 dataset (443 cases) the review period of 2 x 56 days was exceeded 22 times. A possible explanation could be that these were very complex protocols. Unexplainable data (negative timelines (n=7) and SPSS outliers (n=8)) were excluded. 

Information about extension of timelines was added to box 1 (page 4).

2. I note in Table 1 that basically everything get approved! Out of 471 submissions, only 2 were rejected. That is a reject rate of 0.4% -- this is very, very low. Someone can interpret this as a low quality standard -- or, are you sending these submissions back and forth over and over to the PI for corrections, until you give an approval?

First of all it is indeed common practice in the Netherlands that researchers are given the opportunity to answer MREC comments in several comment rounds, adjust protocols and required documents if needed until approval can be given. This process results in low rejection rates. In addition, often, the protocol is first reviewed by a so-called scientific review committee that reviews protocol and scientific quality before submission to the MREC. 

More information about the Dutch situation was added to box 1 (page 4).

3. What the revise and resubmit less or more with the fast-track process? How many cycles in each (regular and fast track)?

We agree that this is an important question and we analysed this for our own MREC.

The METc VUmc approved 6 COVID-19 protocols with an average of 2,0 cycles. 

The METc VUmc approved 34 regular protocols (registered judgement between 13-3-2019 and 20-8-2019) with an average of 2,8 cycles. 

Unfortunately, we do not have national data on the number of cycles in regular and fast track procedures and it is practically not feasible to retrieve this data from all MRECs.

From this data that we have at our disposal, we can conclude that in the regular procedure, on average, slightly more comment rounds took place, but that the difference is not very large. 

4. What about the issue of submissions returned to the PI as they were out of scope (not WMO)? This is known to be a problem in the Netherlands because it is very difficult to determine what is in scope for WMO as the definition is vague and there is the ambiguity of biopsych/psych research. This would apply to COVID as well due to the mental health and isolation issues associated with the topic. Did MRECs find more of this happening with the COVID submissions?

The WMO criteria are indeed partially subject to interpretation. However, all MRECs are used to applying these criteria and many MRECs have an affiliated non WMO committee which makes a careful assessment whether submissions are in scope for WMO. The current research has not indicated that the COVID submissions were assessed differently on this point. In one of the interviews the question was asked: “herkennen jullie de neiging dat je onderzoek eerder als niet-WMO bestempelt in het geval van COVID?”/ “Do you recognize it to label research more likely as ‘not WMO’ in the case of COVID research?” The answer of participant#1 was “Nee dat herken ik niet.”/ “No, I don’t recognize that”. Other participants also did not indicate that they recognize this. 

5. Line 170, are you saying that a digital signature is not permitted (even before COVID)? This seems unusual and burdensome.

We agree with the reviewer that it is a burdensome procedure. However, according to the ‘Algemene Wet Bestuursrecht’ (Dutch General Administrative Law Act) a ‘wet’ signature cover letter is obliged. This obligation is temporarily suspended during the COVID pandemic. Instead, a digital or scanned signature from the submitter is sufficient. As far as we know the future of the ‘wet’ signature is under discussion at the Dutch Ministry of Health. 

6. There is no real explanation of what the fast track process is -- does it mean that it gave a priority to COVID research (over regular research?) or was it that there was an additional reviewing mechanism (a funnel) that took those submissions while there was no change to how regular research submissions were managed?

The Fast Track Review Procedure differed for each MREC. This was added to box 1. The present study was set up to investigate how exactly MRECs developed and implemented their FTRP. Some MRECs set up an ad hoc subcommittee to review COVID-19 submissions accelerated compared to regular research. Some committees also set different requirements for submissions. This was described under ‘Implementation process’ line 139 and further. The overall aim of the FTRPs was to give priority to COVID-19 research over regular research. However, most regular research was put on hold due to COVID-19 measures. Still, 24% of the MREC respondents indicated the review of non COVID-19-related submissions was delayed due to the priority given to COVID-19 research. (line 173/174)

7. I don't see a formal measure of review quality. How was that actually measured?

The reviewer is correct. We also struggled with this question. Review quality is a broad concept, subject to interpretation and worth discussing. We measured the difference in perceived quality by asking if MREC participants found the review quality in general to be different in Fast Track Procedure compared to the regular procedure. In addition, we measured perceived differences in review of specific aspects between the Fast Track Procedures and the regular procedure, e.g. legal, privacy, ethical, methodological aspects (Table 4).

8. What about the issue of the capability of the MREC members to review COVID research? Did you need external experts? How did you find them? Did this impact review time (looking for them)? Or did you have the on-site/internal COVID expertise to review those submissions? This is important to consider in light of the problem with HRECs approving research that is low quality/poor design . See and perhaps reference this paper, https://jme.bmj.com/content/46/12/803

Thank you for referring to this interesting paper. 

However, results of our study show that there was no issue regarding the capability of the MREC members to review COVID-19 research. One of the questions we asked MREC participants was: ‘Did your committee need (more) substantive external support/consultation during the assessment of the SARS-CoV-2 research files?’ 18 of the 21 respondents answered ‘no’, 1 answered ‘yes’ and 2 answered ‘I don’t know’. This was added to line 149. 

Moreover, many hospitals set up a COVID-19 committee that monitored the research burden on the COVID-19 patients and prevented too much (low-quality) research being conducted in this group.

Reviewer #2: 

1. This is a timely and well-written paper on a relevant topic (ethics reviews of COVID research). I have some minor suggestions for improvement:

Thank you for your compliment. 

2. Introduction: A bit more information is required about the functioning of the MREC's in The Netherlands (eg what are typical member's profiles), so that readers can compare these to the working of the MREC's in their own country. 

More information about the member profiles of the MRECs was added to box 1 (page 6). In addition box 1 was supplemented with information requested by reviewer #1. 

3. p6 "As an explanatory follow-up the quantitative results were further explained with in-depth qualitative data"

 'explained' is not really an appropriate term as qualitative and quantitative research have different finality, I would suggest changing it to 'shed more light on'.

Thank you for your suggestion. We changed this sentence to ‘As an explanatory follow-up we conducted group interviews resulting in qualitative data, which shed more light on the quantitative results’ (line 86).

4. p6: how many people were there in each focus group? Best to name exact numbers per group also here. It would be good to provide a table with demographics of focus group participants as well.

Table 1 was added including more information on group interviews and participants.

5. Am I correct that the factual data such as the review times were asked in the questionnaire? So these were based on self-reporting of the participants rather than on actual administrative data? Best to make this clear in the text.

We collected both self-reporting data as administrative data about review times. Because we thought the administrative data were more reliable, the presented review times in table 3 are only administrative data. We made this clear by adding to table 3: ‘(based on data from national registration system CCMO)’ 

6. p10: "Working remotely from home and using video conferences probably contributed to a rapid scheduling of review meetings."  is this what the focus group participants said or is it your own interpretation? Make sure to distinguish this in the results of the qualitative research.

This is what the group interviews participants said. We made this more clear by adding a quotes to line 192 and further. 

7. p13: "The little decline"  the small decline

In line 359 we changed ‘little’ to ‘small’

8. The discussion of the qualitative research is very concise. I feel that this would be helped if some quotes from the transcripts were provided to illustrate the themes.

Quotes were added to the results section of the manuscript to illustrate the themes. Dutch quotes were added as supporting information. 

9. At the end of the paper, there are recommendations in a table, which is fine. However, I would expect a bit more elaboration of these recommendations:  what would these practically entail and how do they relate to your research findings? There is already something in the text preceding the table, but I feel this needs a bit more structure, and the recommendations themselves have to be made more visible in the text.

We agree, thank you for this suggestion. We added elaboration on the recommendations to the discussion section. 

Our study shows what investigators consider important in an (emergency) assessment: in addition to speed, also reachability, a designated contact person, being able to work digitally and distinguishing main and side issues. In addition, the study shows the problems MRECs run into: lack of overview, coordination and cooperation, high workload, legal requirements (wet signature) and (digital) communication options. More research is needed to find out how best to do this and what is feasible or not. To make this more clear we changed the table with Recommendations to a ‘Lessons learned’ box. 

Reviewer #3: 

This paper presents research focused on the experience of researchers and MRECs during the beginning of the COVID-19 pandemic in 2020. The researchers used both administrative data on review length and novel data (both quantitative and qualitative) focused on the experience of reviewers and submitters to understand how the fast track review process (FTRP) impacted COVID-19 research. Overall, the paper does a nice job showing the need for the FTRP, as well as the experience on both sides. The paper should include more technical information, such as more detail about the questionnaire process, any relevant demographics for respondents, how data was analyzed. There are some interesting points in the data that are not reflected in the writing.

Major issues:

1. Please provide more detail about the questionnaire and the semi-structured interview guide; it’s very unclear how they were designed, what was asked, how long they took, what the hypotheses were, etc.

This important comment was also brought up by reviewer one and we refer to the answer given. In addition: we hypothesized review went faster in the Fast Track Review Procedure, but we did not know how much faster and which factors exactly contributed to an accelerated review. It took circa 15 minutes to fill in the questionnaires. Questionnaires were used to quantify the Fast Track Review Procedure nationally and to identify topics that could form a basis for more in-depth evaluation research conform to the principles of Inductive Content Analysis (Kyngäs H. (2020) Inductive Content Analysis. In: Kyngäs H., Mikkonen K., Kääriäinen M. (eds) The Application of Content Analysis in Nursing Science Research. Springer, Cham. https://doi.org/10.1007/978-3-030-30199-6_2). Topic lists were developed based on the most remarkable results from the questionnaires. 

Copies of the questionnaires are included as Supporting Information in both Dutch as English. The topic lists are included in the manuscript as table 2a and 2b. In addition they are included as Supporting Information in both Dutch as English.

2. Some of the more interesting findings are not focused on as much – investigators unanimously thought their research was either equal or better than usual, while reviewers that it was equal or worse (33%!); investigators also seemed to express frustration at legal, tech, participating centers for not moving fast enough. This pokes at a really interesting ethical issue – you have researchers not spending enough time on their own research plans, believing they are as good or better than their other research, and feeling frustrated that legal is concerned…combined with reviewers feeling pressured to approve quickly. All of the recommendations are focused on getting approvals done even more quickly, with nothing focused on the ethics of rushing approvals for protocols that are not adequate.

With regard to the pressure to approve quickly, our data show that MRECs made no concessions with regard to the substantive quality of their review during the FTRP. Priority has been given to COVID-19 research, and MRECs have been lenient on administrative aspects, but our data indicate that leniency regarding administrative matters do not in any way affect the ethical rigour of the review process. Ethical principles that apply to medical scientific research in general equally apply to Covid studies. This finding is in agreement with the position paper of the European network of Research Ethics Committees (EUREC) that concludes: “… the pressure currently being exerted on medical research must not lead to research or testing of pharmaceuticals on humans without complying with the ethical standards applicable to medical research.” Thus, the sense of urgency of Covid research that is widely acknowledged, does not affect accepted ethical and legal standards. 

This was added to the discussion section. 

3. A few references to patients and patient burden – should make clear why patients were not contacted for the study; it would be interesting to at least comment on patient experience here, or discuss as a limitation

We agree it would also be very interesting to study the patient perspective. Contacting patients to participate in this study, however, would not be ethical since it would be a major burden for patients. Moreover, because of privacy restrictions it would be complicated for us as researchers to approach them. Therefore, the patient perspective falls outside the scope of this study. We chose to add this as a limitation to the discussion section (line 427)

Minor issues:

1. 115-118: Provide response rates for individuals, not by MRECs. Provide details on recruitment, follow-ups, compensation

Response rates for individuals are provided. In addition, line 134-135, however, explains how many MRECs are covered by the respondents. 

Details on recruitment were added to the methods section under ‘Data collection’.

In the ethics section was added that no compensation was provided to the participants. 

There are no follow-ups. Recommendations/lessons learned will be discussed in national MREC meetings. 

2. In Implementation Process – it’s confusing to jump between MRECs as entities and individual respondents as entities

Line 126 ‘3 MRECs indicated they consulted another MREC about their FTRP prior to implementation’ was deleted. Now only results from individual respondents as entities were presented. 

3. In total, four semi-structured group interviews were conducted with 10 MRECs representatives and 9 investigators – mixed or by group?

Interviews were conducted by group. Table 1 was added including more information about the group interviews. 

4. A few spelling and grammar errors throughout – nothing major, just recommend a close read to make sure all errors are caught

We checked the manuscript for spelling and grammar errors. 

Reviewer #4: 

Summary of the manuscript: This paper aims to evaluate the fast-track review procedures (FTRP) set up by medical research ethics committees (MREC) in the Netherlands during the COVID-19 pandemic in an effort to meet the urgent need for accelerated review of research proposals. It is an exploratory sequential mixed-method study that uses online questionnaires and in-depth interviews. The authors found that the total number of review days was shorter in FTRP than in regular review procedures (RRP) but this did not seem to impact the quality of review. The main difficulties of the FTRP were the heavy workload for MREC members, the lack of accessibility/coordination between stakeholders, and the lack of “overview of COVID-19 research”. The authors end the paper with a table of recommendations for more efficient and less burdensome FTRPs in the future. Furthermore, in the discussion, the aim is re-phrased so that the goal of the study is to identify differences between FRTP and RRP, rather than to evaluate FRTP in a medical ethical context. Overall, the paper succeeds in giving a description of the FRTP process by MRECs during the COVID-19 pandemic, but not an evaluation or analysis.

Strengths: This paper is ambitious and its objective is very important, as 2020 has been a unique opportunity for examining medical ethical review processes. It also appears methodologically sound. The study received ethical approval and participants provided informed consent. The results are quite striking and thought-provoking, and there is a good, broad summary of the qualitative results.

Weaknesses: The results are straightforward, but lacking in-depth analysis and ethical reflection. It is also unclear what questions were asked in the questionnaire and interviews. The language of the paper is overall moderately difficult to follow, so the paper would benefit from thorough copyediting.

Major issues:

1. It is unclear what questions were asked in the online questionnaire (Line 84-85) and how the interviews were structured (e.g. the topics that were planned to be discussed) and coded (line 99-105) This would be helpful if the authors could include a sample of the questionnaire and the coding, e.g. as part of an appendix/supplementary information.

Copies of the questionnaires are included as Supporting Information in both Dutch as English. The topic lists are included in the manuscript as table 2a and 2b. In addition they are included as Supporting Information in both Dutch as English.

Themes identified were structured on the basis of the three main themes addressed in the interviews: 1. Implementation process 

2. Review timelines and 3. Review experiences. This structure, themes and their number of passages per group were demonstrated in table 5. 

2. The quantitative results section would benefit from a more rigorous statistical analysis, e.g. to identify patterns or trends, and/or differences between investigator findings and MREC findings. Currently, the results are stated without detailed explanation or comparison e.g. Line 143-155

In the results section we mainly show the numbers and percentages. More explanation and elaboration was added to the discussion section. A comparison between investigator findings and MREC findings was also added to the discussion section. 

3. The qualitative results section would benefit from more detailed description of the interview findings. For example, Line 169: explore the “leniency” further. Line 179-183: How much impact did the delay of regular research have on the investigators? It would further be important to analyse the ethical implications of delaying appraisals of regular research, perhaps in the discussion section.

To give more explanation of the qualitative results we added quotes to the results section. Dutch quotes were added as supporting information. 

We agree it is also very interesting to analyse the (ethical) implications of delaying appraisals of regular research. However, we feel this is beyond the scope of our investigation as we only have data from COVID-19 investigators. To investigate this also data from investigators whose regular investigation has been delayed is needed. Since it is interesting to investigate this, we have added this to the discussion section as a suggestion for further research. 

4. The discussion section seems to re-state the results. It would benefit from an in-depth ethical analysis, e.g. the implications of “leniency” of required documents, or the increased need for human tissues (line 211-212.) The paper would also benefit from a discussion of the strengths/weaknesses of the study, e.g. sample size, subjective nature of the data (like results mentioned in Line 194-195), and applicability of the findings.

More explanation and elaboration on the results and recommendations was added to the discussion section. Furthermore, a limitations section was added discussing the weaknesses of the study. 

5. The discussion section would benefit from a description of the recommendations before outlining them in a table, as well as giving examples and evaluating the benefits, costs, feasibility, and justification of the recommendations.

More explanation and elaboration on the recommendations was added to the discussion section. It is not our intention to suggest to simply adopt our recommendations as new policy. More research is needed to determine which recommendations are feasible and justified. Therefore, the table with Recommendations was changed to a ‘Lessons learned’ box. 

6. While some of the findings are endorsed by previous papers (Line 260) the points in discussion section need to be tied back into the existing literature to place the findings in a wider context, especially when analysing the ethical implications or providing recommendations.

The discussion sections was modified based on this point. For example, we added the position of the European network of Research Ethics Committees (EUREC).

7. Overall, the language, sentence structure, and (lack of) punctuation decrease the readability of this manuscript. This also makes some of the results unclear. For example: line 210-211 - the protocols did not have a “good overview of the availability of patients and possibilities in the hospital”?

We checked the manuscript for spelling and grammar errors. 

Line 210-211 was changed to: ‘they indicated having not always an accurate overview of the availability of patients and competing interests of different research questions’ (line 309).

Minor issues:

8. The tables are not very easy to read; they could benefit from more formatting

Since this is a minor issue and we expect PLOS ONE to also edit the tables we chose not to change the formatting of the tables. 

Reviewer #5: 

This manuscripts reports the results of a mixed method student into the fast track reviewing procedures that medical research ethics committees (MRECs) in the Netherlands implemented during the COVID-19 pandemic in 2020. Overall, the research design is sound and the data reported supports the conclusions, and the results are of interest to an international audience as human research ethics committees worldwide have implemented similar fast track review procedures throughout the pandemic. However, several issues need attending to bring the manuscript up to a standard applicable for an international journal and audience. I have made many suggestions in the comments of the PDF manuscript file (attached). Notably, the authors should append into the Supp Materials a completed COREQ checklist for qualitative research as well as the the survey questions and interview guide. In the manuscript they should provide an assessment of response/compliance rates and address some discrepancies in responses shown in Table 2.

- The suggestions in the PDF manuscript file have largely been adopted. 

- The COREQ checklist was used and added as Supporting Information.

- The topic lists are included in the manuscript as table 2a and 2b. In addition they are included as Supporting Information in both Dutch as English.

- Response rates were discussed in the limitations section added to line 422

- Numbers in Table 4 should not add up to the questionnaire samples because per aspect it is indicated how many respondents indicated a difference. 

Substantively, while I concur with the authors recommendations for improving the fast track procedures of MRECs, it is unclear why they could also not be for regular review processes, as it seems that this system is in need of improvement as well – 90 days for regular review seems excessive and a national approach may help to reduce duplication. I’m unsure if an exceptional case can be made – why not just improve the whole review system while we are here?

We agree with the reviewer that our recommendations largely also hold for the RRP. However, more research is needed to determine which recommendations are feasible. The FTRP cannot simply be taken over during normal circumstances. It is also not our intention to suggest to simply adopt the recommendations as new policy. Therefore, table 3 Recommendations was changed to a ‘Lessons learned’ box and suggestions for further research were added to the discussion section (lines 413-417 and lines 427-430). 

Finally, the manuscript contains some awkward phrasing throughout and should be thoroughly edited for an English-language international journal.

We checked the manuscript for spelling and grammar errors. 

Reviewer #6: 

Introduction

1. It is not clear why the information about the Dutch setting has been put in a box. I would suggest this material is simply described in the introduction section. And that Dutch names of legislation are excluded.

Information about the Dutch setting is background information and has been put in a box to keep the introduction section short and to the point. Moreover, reviewer 1 and 2 ask for more information about the Dutch situation. Therefore, we chose to keep the box.

However, we tried to keep the information in the box as short as possible. Dutch names of legislation are excluded from box 1.

2. Although the literature of ethics review of research in pandemics may be limited, it would be helpful if the authors outline the key findings of this research and WHO recommendations in more detail in the introduction section, including the few publications regarding ethics review in the current pandemic.

Key findings of previous literature was added to the introduction section line 61. In the discussion section a reference to the EUREC position regarding ethics review in the current pandemic was added. 

3. The aims and importance of the study are not clearly stated at the end of the introduction section

Aims and importance of the study are more clearly stated in the introduction section now (lines 70-75)

Methods

4. The methods are currently described in insufficient detail. More details are needed regarding the development, contents, implementation, and analysis of the survey, and regarding the design, implementation, and analysis of the group interviews. It would be preferable if the authors followed reporting guidelines.

The method section has been expanded with information regarding the questionnaires and group interviews. 

Copies of the questionnaires are included as Supporting Information in both Dutch as English. The topic lists are included in the manuscript as table 2a and 2b. In addition they are included as Supporting Information in both Dutch as English.

The COREQ checklist was used and added as Supporting Information. 

Results

5. Quantitative results section is difficult to follow and I would suggest that authors revise to improve clarity and readability. It would be helpful is the authors provide demoniators numerators, and a table provide full results.

We complied with the English convention to alphabetical spell out numbers at or below 10 and that start a sentence, suggested by reviewer #5. Demoniators numerators were used in the rest of the results section.

6. The authors should consider adding a table that provides an overview of the interview codes and subcodes identified.

Code structure, themes and their number of passages were demonstrated in table 5.

Discussion

7. The discussion section should begin with a clear statement of the key findings of the study. Currently, there is too much repetition of study aims and results.

This has been adjusted at the beginning of the discussion section (lines 365-366). 

8. The discussion of the results in the context of existing literature is not sufficient, partly reflected by the article citing a total of 11 publications. Although there may be limited reearch focusing on ethics review of research in pandemics, there is certainly relevant research that the authors should draw on.

We have adopted this advice and added a comparison of our findings with the position paper of the European network of Research Ethics Committees (EUREC) (lines 374-379).

9. The practical implications of the research need to be discussed in more detail. Recommendations are listed in table 3, but the authors should discuss these in the discussion section. As these recommendations are from the authors, the summary should also be in a box and not a table.

More explanation and elaboration on the recommendations was added to the discussion section (line x). It is not our intention to suggest to simply adopt our recommendations as new policy. More research is needed to find out how best to do this and what is feasible or not. To make this more clear we changed table 3 Recommendations to a ‘Lessons learned’ box x and suggestions for further research were added to the discussion section. 

10. The limitations of the study are not described.

A limitation section was added to the discussion section (lines 419-430).

Reviewer #7: 

In this mixed-methods study, the authors evaluate how expedited research review in the Netherlands during the COVID-19 pandemic affected processes and perceptions form both investigators and board review members. This is one of the first studies of its kind to use mixed-methods techniques to study implementation of expedited research review during COVID, and as such is likely to be of significant interest to the research community. The generalizability of this study is questionable given that nation-wide efforts expedited review processes were rare in countries like the US, where research review is usually performed by single institutions. However, as these institutions likely had COVID fast track policies, there are still learning lessons.

A few comments:

INTRODUCTION

1. Some further detail on the FTRP in the introduction could be helpful. Did the FTRP apply only to COVID-related research? Did it apply only to prospective studies, or retrospective/observational studies as well? Furthermore, was other non-COVID research allowed to be submitted for review during the COVID period? Did these policies vary by institution or were they set country-wide?

The FTRP only applied to COVID-19-related research. 

16/21 MREC respondents having a FTRP indicated the FTRP applied to both research subject to the Medical Research involving Human Subjects Act (prospective studies) as other research, for example retrospective studies. 5/21 MREC respondents having a FTRP indicated the FTRP only applied to research subject to the Medical Research involving Human Subjects Act. 

 The CCMO published an ‘advice for the conduct of clinical research at the time of the restrictive measures by the coronavirus’ (version 23 february 2020) which states: “As a sponsor or researcher you should ask yourself whether the research, or part of the research, can temporary be put on hold or not.” In addition, organizations also had their own policy. This meant some studies were temporarily halted and others continued. It was allowed to submit non-COVID research. 

METHODS

My biggest issues apply to the qualitative portion

2. The authors should follow a published checklist, such as the COREQ checklist, for reporting their qualitative study.

The COREQ checklist was used and added as Supporting Information. 

3. The authors should describe how saturation of interview themes during the interview period was arrived at.

We invited all participants who were willing to participate in a group interview, resulting in 4 groups interviews (see table 1). During the second group interview with MRECs as well as investigators we concluded that no new topics were raised and that saturation had thus been achieved. This was added to line 103 and 423. 

4. In the methods, The number of interview guide questions should be included, even though the guide itself is provided as a supplement.

The topic lists and interview questions are included in the manuscript as table 2a and 2b. In addition they are included as 

Supporting Information in both Dutch as English.

5. A weakness in this paper is the lack of a cited theoretical model or foundation for the qualitative interviews. How were codes arrived at for the qualitative portion? Was a grounded theory approach, or another method used?

The principles of Inductive Content Analysis were followed. 

Information was added to the manuscript (line 83). 

6. To calculate review timeline, were institutions required to send all of their COVID reviews for analysis, or was there some sort of convenience sampling where institutions were allowed to choose which COVID reviews to send for analysis?

Median review timelines were calculated using data from the national registration system of the CCMO in which all MRECs in the Netherlands report their review periods. In table 3 ‘Based on data from the national registration system CCMO’ was added. To line 420 and further was added: ‘The administrative data used to calculate review timelines concerned data entered to the CCMO database by MRECs themselves. 

RESULTS

7. The manuscript would significantly benefit from inclusion of direct quotes from interview transcripts to support some of the general points made. Alternatively, a table of representative quotes within each theme would be beneficial.

Quotes were added to the results section of the manuscript to illustrate the themes. Dutch quotes were added as supporting information. 

8. How were the number of 4 total interview groups arrived at? How many participates participated in each group?

We invited all questionnaire respondents who were willing to participate in a group interview, resulting in 4 groups interviews (see table 1).

Reviewer #8: 

1. I would first like to note the importance of this topic and the potential impact of this research. I especially respect the integration of qualitative data here, and I appreciate this research team’s identification and exploration of such an important issue.

Thank you for your compliment.

2. I’m concerned that the level of detail provided in the Methods section is insufficient to understand the data; in particular, expanding the description of the instrumentation would be helpful to understand the meaning of the data; e.g., how were these surveys/interview guides developed? Was there cognitive interviewing or pilot testing? Were they based on any validated instruments, literature, etc.? What data was collected via survey versus secondary analysis of CCMO records, and how were these integrated?

Since this topic concerns a unique emergency situation, questionnaires were primarily developed on basis of the experience with procedures of our own MREC (METc VUmc). Questionnaires were used to quantify the Fast Track Review Procedure nationally and to identify topics that could form a basis for more in-depth evaluation research conform the principles of Inductive Content Analysis.

Information was added to the manuscript (Design page 6). 

We collected both self-reporting data via the questionnaires as administrative data from the CCMO databank about review times. Because we thought the administrative data were more reliable, the presented review times in table 3 are only administrative data. We made this clear by adding to table 3: ‘(data from national registration system CCMO)’ 

3. Further, additional explanation and description of the coding and thematic analysis would be helpful to understand the level of rigor involved in the methodology; e.g., did the coders measure any sort of inter-coder agreement? I do see that the coders “independently” applied the codebook to the transcripts, but am not clear on whether they each coded all 4 transcripts in their entireties. And were these a priori structural themes? Emergent thematic?

Two researchers independently coded all 4 transcripts in their entireties after which coding was discussed and a definitive coding structure was determined. Agreements were made about when a code should be given to a passage, for example that when a particular topic was discussed, the entire passage in the transcripts was coded with one code. Afterwards, all transcripts were coded based on the agreements made. All researchers agreed with the final encoded transcripts. 

This explanation was added to Data analysis (page 8 and 9). 

4. At several points throughout, there was some confusion re: the extent to which statements were actual presentations of data, versus conjecture/observation/opinion of the authors; e.g., “More central management and collaboration between different research groups and MRECs could improve this.” [216-7] -- it is unclear if this is an actual finding emerging from the data or just an argument from the authors. There are several similar instances throughout.

We adjusted the results section to make more clarify which statements were study results, for example by adding quotes. 

5. The presentation of quantitative results as separate from qualitative results—and the descriptor of “mixed method study”—is confusing. According to the authors’ statement that “as an explanatory follow-up the quantitative results were further explained with in-depth qualitative data” [81-2], it seems more informative/appropriate to present these results in a cohesive, synthesized analysis. Some qualitative researchers would argue that collecting follow-up qual data from participants to further inform their prior quant responses does not constitute a “mixed methods” study design. I don’t have a strong opinion on this, but I wanted to point this out so the authors can revise to clarify or preemptively defend against potential critiques/confusion.

We followed Creswell et al who state that an explanatory sequential mixed methods design ‘follows the form of first reporting the quantitative, first-phase results and then the qualitative second-phase results’ Then ‘a discussion should follow that specifies how the qualitative results help to expand or explain the quantitative results’. 

6. I’m somewhat concerned about the framing of investigators being “satisfied” with the review process (e.g., [143-4])—investigator satisfaction is not the goal of any ethics review in any circumstances, and should not be messaged as a measure of quality, success, or efficiency.

We agree that satisfaction should not be the goal of ethics review. Also, it is not a measure of quality. However, it does give an indication of how people view how this procedure went. 

7. I’m also struck by the reporting of quant/qual data re: MERC reps’ quality assessment of reviews, only insofar as a MERC rep could reasonably have patent or latent bias in that assessment; i.e., if standard review is the minimum threshold for ethics review, why would a MERC rep ever consider their review to be below the ethical standard, much less discuss that openly with other MERC reps? For the sake of clarity: theoretically, the MERC reps who reviewed these studies would *by definition* believe that their review was appropriate, otherwise they would not have done it that way. Further, given the method chosen here (group interviews), a rep who questioned the quality of a review would have little-to-no incentive to disclose that doubt amongst their peers. But perhaps I’m missing something here.

We agree that it makes sense to assume MRECs find their review to be of good quality. However, we hypothesized that the urgency was highly taken into account in the FTRP. Therefore we asked MRECs to compare their review quality to the regular situation. It was possible that the MRECs had opted for a slightly lower (but still appropriate) quality of review in view of the emergency situation. 

Regarding the method chosen (group interviews): we have tried to emphasize confidentiality and there were small groups. Moreover, everyone had an interest in properly evaluating this procedure to improve review in the future. We also refer to critical remarks made by MREC representatives (e.g. quote line 297) Therefore, we think this method was appropriate, although we agree it cannot be ruled out that participants were not completely open about their doubts. 

8. Overall, I worry that the authors may have limited the value/applicability/impact of their study by presenting what appears to be a cursory analysis, with the results section comprising little more than a recitation of (presumably structural) themes. Additional themes, further exploration, or just a more extensive analysis would be helpful to better support the study’s overall impact.

More explanation and elaboration on the results and recommendations was added to the discussion section (line x). Furthermore, a limitations section was added discussing the weaknesses of the study. More research is needed to find out how best to apply the recommendations and what is feasible or not. To make this more clear we changed table 3 Recommendations to a ‘Lessons learned’ box and suggestions for further research were added to the discussion section. 

9. The discussion of other review timelines seems out of place and uninformative considering the wide range of legal/regulatory requirements in oversight/review, additional systemic burdens from the pandemic, etc.

We agree. These sentences were deleted. 

10. It would be helpful to provide some amount of qualitative data to make sense of the themes presented; without this, readers cannot know how salient a theme was, if 2 or more themes were from the same groups/participants, etc.

Themes and their number of passages were demonstrated in table 5. 

11. “These changes did not affect the quality of the review. Moreover, the FTRP is associated with a high degree of satisfaction among both investigators as well as MREC members.” [252-3] Is this first statement demonstrably true based on the data? Or is this an argument by the authors? There doesn’t seem to have been any actual measurement of quality beyond the perceptions of those who were (i) being reviewed or (ii) doing the very review of which the quality is in question.

The point of measuring quality was also addressed by reviewer #1 and we also struggled with this point. Review quality is a broad concept, subject to interpretation and worth discussing. This makes it complicated to properly measure quality. We measured the difference in perceived quality by asking if MREC participants found the review quality in general to be different in Fast Track Procedure compared to the regular procedure. In addition, we measured perceived differences in review of specific aspects between the Fast Track Procedures and the regular procedure, e.g. legal, privacy, ethical, methodological aspects (Table 4).

The statement in line 252-3 was based on for example the questionnaire data which show that ‘90% of the MREC representatives indicated the quality of their review was equal to regular review’. 

12. The overall framing of the results and discussion is focused on investigator and MERC rep perspectives, with little-to-no acknowledgment of competing priorities (e.g., clinical care, allocation of limited resources); i.e., to recommend any action/policy without representation of additional stakeholders seems unwise at best.

Though our data show that a rapid review is possible without compromising on quality, ethics or law, it is indeed important to consider the special circumstances. The FTRP cannot simply be taken over during normal circumstances. It is not our intention to suggest to simply adopt the recommendations as new policy. More research is needed to determine which recommendations are feasible. We added this to the discussion section and Table 3 Recommendations was changed to a ‘Lessons learned’ box. 

13. Some attention is needed for minor grammatical and typographical errors.

We checked the manuscript for spelling and grammar errors.

---

## [Decision Letter · Decision Letter 1]

9 Jul 2021

Medical ethical review of COVID-19 research in the Netherlands; a mixed-method evaluation among Medical Research Ethics Committees and investigators

PONE-D-21-05904R1

Dear Dr. IJkema,

We’re pleased to inform you that your manuscript has been judged scientifically suitable for publication and will be formally accepted for publication once it meets all outstanding technical requirements.

Kind regards,

Prof. Ritesh G. Menezes, M.B.B.S., M.D., Diplomate N.B.

Academic Editor

PLOS ONE

Additional Editor Comments:

- Title: Replace ''Medical ethical review'' with ''Ethical review''.

Reviewers' comments:

Reviewer's Responses to Questions

**Comments to the Author**

1. If the authors have adequately addressed your comments raised in a previous round of review and you feel that this manuscript is now acceptable for publication, you may indicate that here to bypass the “Comments to the Author” section, enter your conflict of interest statement in the “Confidential to Editor” section, and submit your "Accept" recommendation.

Reviewer #1: All comments have been addressed

Reviewer #2: All comments have been addressed

Reviewer #4: All comments have been addressed

Reviewer #6: All comments have been addressed

Reviewer #7: All comments have been addressed

2. Is the manuscript technically sound, and do the data support the conclusions?

Reviewer #1: Yes

Reviewer #2: Yes

Reviewer #4: Yes

Reviewer #6: (No Response)

Reviewer #7: Yes

3. Has the statistical analysis been performed appropriately and rigorously? 

Reviewer #1: Yes

Reviewer #2: Yes

Reviewer #4: I Don't Know

Reviewer #6: (No Response)

Reviewer #7: Yes

4. Have the authors made all data underlying the findings in their manuscript fully available?

Reviewer #1: Yes

Reviewer #2: Yes

Reviewer #4: Yes

Reviewer #6: (No Response)

Reviewer #7: Yes

5. Is the manuscript presented in an intelligible fashion and written in standard English?

Reviewer #1: Yes

Reviewer #2: Yes

Reviewer #4: Yes

Reviewer #6: (No Response)

Reviewer #7: Yes

6. Review Comments to the Author

Reviewer #1: great work to satisfy 8 peer reviewers! best of luck with your continued research -- this is a very important topic.

Reviewer #2: I am satisfied with the changes that the authors have made, it is now ready for publication. Thank you.

Reviewer #4: My previous comments have been addressed. However the paper would benefit from more copy-editing to address minor issues.

Reviewer #6: (No Response)

Reviewer #7: The authors have sufficiently responded to my critiques. This paper is likely to be of significant relevance to the international research community.

7. PLOS authors have the option to publish the peer review history of their article (what does this mean?). If published, this will include your full peer review and any attached files.

Reviewer #1: No

Reviewer #2: **Yes: **Kristien Hens

Reviewer #4: No

Reviewer #6: No

Reviewer #7: No

---

## [Editor Report · Acceptance letter]

15 Jul 2021

PONE-D-21-05904R1 

Ethical review of COVID-19 research in the Netherlands; a mixed-method evaluation among Medical Research Ethics Committees and investigators 

Dear Dr. IJkema:

I'm pleased to inform you that your manuscript has been deemed suitable for publication in PLOS ONE. Congratulations! Your manuscript is now with our production department. 

Kind regards, 

on behalf of

Prof. Dr. Ritesh G. Menezes 

Academic Editor

PLOS ONE